# Spatial Dynamics of the Shore Coverage within the Zone of Influence of the Chambo River, Central Ecuador

**Julie Echeverría-Puertas** [1], **Magdy Echeverría** [2], **Franklin Cargua** [1] **and Theofilos Toulkeridis** [3,*]

1   Escuela Superior Politécnica de Chimborazo (ESPOCH), Grupo de Investigación y Desarrollo para el Ambiente y Cambio Climático (GIDAC), 060101 Riobamba, Ecuador
2   Escuela Superior Politécnica de Chimborazo (ESPOCH), 060101 Riobamba, Ecuador
3   Departamento de Ciencias de la Tierra y de la Construcción, Universidad de las Fuerzas Armadas ESPE, Sangolquí 171103, Ecuador
*   Correspondence: ttoulkeridis@espe.edu.ec; Tel.: +593-987001807

**Abstract:** The predominant aim of the current study was to evaluate the spatial dynamics of the riparian coverage of the area of influence of the Chambo River in the area of the river's source (middle-high basin), between 2500 and 3000 m.a.s.l. For its execution, Landsat 7 images from the year 2000, RapidEye from the year 2009, and Spot 6 from the year 2019 were used in the time range of 2000–2009 and 2009–2019. These were subjected to supervised classification by applying the maximum likelihood algorithm, identifying five classes of soil cover, being pasture, crops, soil-remnants of paramo, forest, and anthropic. The classification results were validated by calculating the precision measures and the kappa index. With the use of cross-tabulation matrices, the gains, losses, and persistence in the two periods studied were identified. There, it was determined that, in the first study period, the soil cover-paramo remnants presented the highest percentage of loss (26.70%), the crop cover the highest percentage of gain (28.91%), and in the second period, the crop class presented the highest percentages of losses (18.94%) and gains (17.29%). The cartographic projection of the area for the year 2030 predicts that the areas anthropic category will increase by 1.27%, that of forest will decrease by 1.19%, that of soil-remnants of paramo will gain 0.79%, and crop and pasture cover will decrease by 0.45% and 0.43%, respectively. The results obtained allow for the transitions between coverages to be attributed to population growth, afforestation, reforestation, deforestation and agricultural activities, volcanic eruptions, land colonization, and expansion of agricultural activity. Complementary studies are recommended that involve livelihoods and water quality, which facilitate the identification of vulnerable areas to propose adaptation, prevention, and/or restoration measures.

**Keywords:** Chambo River; Andean riparian area; spatial dynamics; land cover; supervised classification; cartographic projection



## 1. Introduction

Riparian ecosystems are characterized as ecologically complex systems because they constitute transition zones or interface, as it is an ecotone located between terrestrial and aquatic zones [1–5]. Its importance is reflected in the multiple interactions that occur between aquatic and terrestrial ecosystems, in the biological connections that provide, between plants and animals, species dispersal corridors and the dynamics of interaction between different species [6,7]. Thus, any alteration at the basin level can affect the natural balance of water bodies [8]. The regulation of the flow, the introduction of exotic species, and the change of land cover determine the alteration of riparian ecosystems, causing important impacts [9–11]. In this way, in the area of influence of a river, loss of land cover can be identified due to the fragmentation of the land in order to conduct the construction of houses, industries, and productive plots [12–16].

Hereby, it is essential to consider that the surface characteristics of the ecosystems regulate the radiation balance and the mass and energy flows between the soil and the atmosphere, the intensity of the wind with its roughness, and the humidity of the superficial layer. Thus, land use, an activity characterized by the type of cover, particularly plant cover, plays an important role in regulating the climate and different phases of the hydrological cycle because the surface characteristics of ecosystems regulate the radiation balance and the mass and energy flows between the soil and the atmosphere, the intensity of the wind with its roughness, and the humidity of the surface layer [17–19]. In this way, changes in land cover contribute to a series of impacts such as climate change, which is not only a consequence, but can also be one of the causes of changes in cover, accompanied by high levels of erosion, reduced plant cover, loss of productive capacity of the soil, high rates of migration, and decreased quality of life in rural communities due to inadequate agricultural management [20–25].

The changes in land cover riparian zones with the subsequent loss of native plant cover cause the fragmentation of habitats and ecosystems [8,26], which negatively affects ecological processes and endangers the species of flora and fauna associated with the place in such a way that its conservation and restoration is an essential issue to address when considering the importance of these ecosystems [27–30]. To accomplish this, an analysis of the spatial dynamics of the study area will allow for quantification of the changes in land cover, and thus the formulation of conservation strategies [20,31,32].

The riverside ecosystems in Ecuador are indicators of the resilience capacity because they receive all the discharge from the activities in the hydrographic basin. In this context, if the dynamics of the coverage change are known, we will be able to define a tolerance limit of riverside ecosystems to propose solution strategies in the medium- and long-term [33–38]

The Technical Secretariat of the Management Committee of the Chambo River sub-basin in 2013 reports that this sub-basin covers 164,974 hectares of moors and forests, which have been lost in great quantity since 2000. This is the case of the moors in the upper part of the basin, which support the advance of the agricultural frontier, reforestation with exotic species, burning of pasturelands in grazing areas, and the main livestock and sheep activities [39–41]. This causes much destruction, in addition to the migration of fauna, the loss of water sources, and the reduction in the flow of rivers because the paramos play a very important role in the water cycle [42]. Thus, spatial dynamics studies provide information that will allow for the development of technologies or plans for adaptation and mitigation to this problem, taking into account the importance of the Andean riparian areas due to the characteristic agricultural and livestock activity practiced for centuries and the presence of paramo ecosystems that are very important due to their carbon storage capacity and for being the largest provider of water of excellent quality in the Andean highlands [17,43–45].

Based on the aforementioned, the predominant aim of the current study was to evaluate the spatial dynamics of the riparian coverage of the area of influence of the Chambo River. Thus, it will be possible to stratify the riparian coverage in the area of influence of the source of the Chambo River, within the upper middle basin between 2500 and 3000 m.a.s.l. Additionally, we intend to analyze the coverage present in the study area in the years 2000, 2009, and 2019 using remote sensors to cartographically determine the projection of the present coverage in order to project its development up to the year 2030.

## 2. Materials and Methods

### 2.1. Study Area

The study area is situated in central Ecuador in the Interandean Valley or Depression (Figure 1) [46,47] within the Upper Middle Basin of the Chambo River between 2500 and 3000 m above sea level, which corresponds to sections of the main tributaries, being Guamote, Cebadas, and Alao.

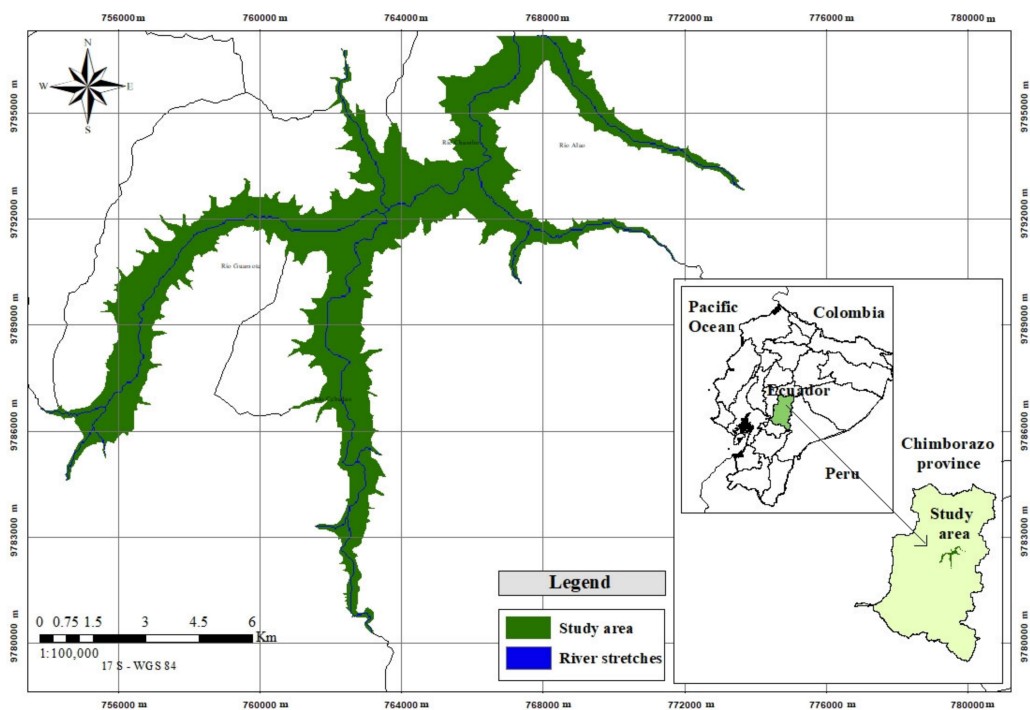

**Figure 1.** Location map of the study area.

For the first time, a water network formed by branches greater than and equal to three was identified, in order to distinguish the main tributaries within the Chambo River sub-basin. To accomplish this, using the GIS software, several steps were performed [48].

For the identification of the water network, we started with a tool that corrects the digital elevation model (DEM) raster by filling in the empty pixels, which was obtained from the U.S. Geological Survey Earth Explorer at 30 m resolution. Then, the direction information of flow from the DEM was obtained. In the same way, the accumulated flow for each cell was calculated [49]. Next, the flow accumulation raster was used in order to calculate the stream raster that indicates information from the drainage network. Subsequently, it was also converted into continuous segments to the drainage network using the raster of the drainage network and direction of flow. To create a raster of the order of the drainage based on the Strahler method [50], the raster generated with the previous tool was applied. In the last step, a drainage shapefile was generated from the order of the drainage and the flow direction, obtaining a shapefile of the water network. Analyses were conducted in ArcGIS.

For the delimitation of the hydrographic unit of the Chambo River, the contour lines and hydrography were obtained in the shapefile format from the Military Geographic Institute (IGM) web portal. Furthermore, the delimitation of the study area was realized through a stratification based on altitude ranges, where the area of interest was located in a range between 2500 and 3000 m above sea level using the corresponding DEM data, managing to determine a net area of 4669.42 hectares around the point of origin of the Chambo River. Thus, we obtained a fully defined area within the upper middle basin of the Chambo River. The focus of this investigation was the area associated with the study area; from this value, the control points of the investigation were determined using a completely random design and Fisher's formula for the calculation of samples of finite populations [51], defining 355 control points.

### 2.2. Pre-Processing of Satellite Images

The Landsat 7, RapidEye, and Spot 6 satellite images were also used. The Landsat image was obtained from the United States Geological Survey Earth Explorer website, considering cloud cover less than 30% so that the visible field is as wide as possible, while

the RapidEye and Spot images were provided by the PACHA project of the research group that supports the present study. The selection of the three types of satellite images was carried out in order to have the widest visible field possible in each of the years of study. The spatial resolution of the images was: 30 m for Landsat 7, 5 m for RapidEye, and 1–2 m for Spot 6.

The pre-processing was only applied to the Landsat image, since the RapidEye and Spot images were provided with the respective correction. The pre-processing of the Landsat 7 image implies an atmospheric correction, which requires a prior radiometric calibration that converts the digital number (ND) of each band into a radiance value (L), and to convert the radiance to reflectivity values in the top of atmosphere (TOA) [50]. In order to convert the ND values to radiance values, the radiometric rescaling factors G and B are used, which are specific for each band [52], and are supplied by the MTL metadata file. Subsequently, the calculated spectral radiance is converted into a planetary or exoatmospheric reflectance (TOA) [53]. In this way, the atmospheric correction is performed, using the dark subtraction method of the ENVI Software [54].

*2.3. Classification Analyses*

In the three satellite images used in this study, the combination of spectral bands in false-color was applied, that is, the three bands (blue, green and red) that correspond to the visible spectrum. This combination is the closest to the appreciation of the Earth with our eyes from space, in other words, a false color image is used to reveal or enhance features that would otherwise be invisible or poorly visible to the human eye [55]. In this phase, a supervised classification was conducted, in order to obtain a land cover map of the study area using the "Semi-Automatic Classification" library in the QGIS software. For this, it is necessary to establish control points, which refer to the areas or regions of interest (ROIs), also called training plots [56]. The 355 control points were identified in the field with the use of a GPS, a sample value that was calculated based on the study area hectares. The identification of these points was carried out randomly around the entire area, equally identifying the number of points in the five land cover classes. Later, they were located in the satellite images, adding the identified coverages with the "Macro Class List" tool, and then using the icon to create a polygon ROI [57].

For the methodology assignment phase, the maximum likelihood classifier was selected, also using the QGIS software, as it is a simple algorithm to apply and is considered as one of the most accurate and efficient discrimination procedures [58]. This algorithm produces the "spectral portrait" of each category based on the mean and variance of the training sites located in the satellite image. The data are considered to follow a normal distribution, which allows for working with a probability distribution model that allows each pixel to be assigned to the category with the highest probability of belonging [59]. On the other hand, the stratification of the study area consisted of identifying the present soil cover such as pasture, crops, soil and remnants of paramo, forest, and anthropic, which refers to constructions, roads, and any activity in which man intervenes, except the agricultural part that is already included in the other designated coverages.

The area delimited for the study is located in the upper middle basin of the Chambo River, where the upper zone corresponds to the Andean moorland and the lower zone to the productive part. The characteristic covers are pasture and Polylepis forests (paramo areas) [60], eucalyptus (Eucalyptus globulus), and pine (Pinus radiata) [61]. The latter is the main cause of changes in coverage, since they are introduced species that have the objective of increasing or diversifying the economic income of Andean peasants, reducing the erosion of degraded areas, and also as a carbon sequestration strategy [62]. Agricultural plots and anthropic constructions are coverages that have also been expanded due to the population increase.

The study area, located in the upper middle zone of the Chambo River basin, is composed only of the sections of the main rivers (Chambo, Guamote, Cebadas, and Alao) and their tributaries. The section of the Cebadas River (order 3) that was part of the study area is 17,148.14 m long and is delimited by the altitude of the 2880–3040 interval; the 16,163.84 m of the section of the Alao River (order 2) is delimited by 2760–3040; the section of the Guamote River (order 4) involved is 13,918 m located between 2840–3000; and the 9782.13 m section of the Chambo River (order 4) is located between 2760–2840. Then, the study area was stratified according to the types of coverage identified, considering the coverage of interest for the MAE, presented in its thematic legend, of levels I and II [63,64]:

- Pasture (Level II): Vegetation characteristic of the slopes of the mountain ranges, belongs to the agricultural land group (Level I) of the coverage classification.
- Crops (Level II): Coverage belonging to agricultural land. Annual, semi-permanent, and permanent crops can be identified in the Chambo River sub-basin, with corn and rice the predominant ones.
- Forest: Coverage belonging to Level I of the coverage classification, which includes forest plantations and native forest (Level II). The forest cover present in this study area is composed of Polylepis forests (native), species of Eucalyptus globulus and Pinus radiata.
- Anthropic Zone (Level I): Refers to the populated areas found in the sub-basin. Therefore, it refers to constructions and roads belonging to Level II of the afore-mentioned classification.
- Soil and remnants of paramo: This refers to land with bare soil and remnants of vegetation belonging to paramo areas. They were not selected as separate covers because the existing density of paramo remnants is not significant.

The supervised classification was validated to analyze the land covers identified in the years 2000, 2009, and 2019 by applying an error matrix that allows for calculation of the precision measures and the Kappa index. The error matrix is a square matrix, and the rows and columns that make it up represent each of the categories of the land cover map obtained from the performed classification [65].

Within the precision measures, there is the Producer's Accuracy (PA), which refers to the probability that a given class is correctly recognized, the User's Accuracy (UA), which represents the probability that a pixel is correctly classified in the specific class to which it belongs, and the Overall Accuracy (OA), which indicates the proportion of the area correctly classified [66]. As for the Kappa index, it is a measure of the concordance between two maps considering all elements of the error matrix [67]. Consequently, the multi-temporal evaluation was directed to the identification of gains, losses, and persistence of each of the land covers. Therefore, a coverage change transitional matrix was used. This is a cross-tabulation matrix valued as an ideal tool for the descriptive analysis of transitions [66]. In this way, the changes and vulnerability measures were calculated, as listed in Table 1.

The cross tabulation matrix is also known as the change matrix in land use studies. It is composed of rows that have the information of the classes of time 1 and the columns of time 2 [67]. The parameters for the analysis of land cover changes determine the gains, losses, net change, total change, and exchanges for each class (cover) experienced between the two study times, and the total change is half of the sum of the changes in the individual categories. In the same way, it happens with the net change and the exchange, as the sum of the changes in the individual categories counts twice the change in the total area because the change in one cell of the matrix counts as a gain in one category and a loss in another category [68]. In addition, the Braimoh persistence indices relate the persistence of the loss and gain of each category [69].

**Table 1.** Measures of the changes and vulnerability of the study area.

| | | | Cross Tabulation Matrix | | | | |
| --- | --- | --- | --- | --- | --- | --- | --- |
| | | | Year 2 | | | | |
| **Year 1** | **Pasture (1)** | **Crops (2)** | **Forest (3)** | **Anthropic Zone (4)** | **Soil and Remnants of Paramo (5)** | **Total Year 1** | **Losses** |
| Pasture (1) | $A_{11}$ | $A_{12}$ | $A_{13}$ | $A_{14}$ | $A_{15}$ | $\Sigma A$ | $\Sigma A - A_{55}$ |
| Crops (2) | $A_{21}$ | $A_{22}$ | $A_{23}$ | $A_{14}$ | $A_{15}$ | $\Sigma A$ | $\Sigma A - A_{11}$ |
| Forest (3) | $A_{31}$ | $A_{32}$ | $A_{33}$ | $A_{34}$ | $A_{35}$ | $\Sigma A$ | $\Sigma A - A_{33}$ |
| Anthropic Zone (4) | $A_{41}$ | $A_{42}$ | $A_{43}$ | $A_{44}$ | $A_{45}$ | $\Sigma A$ | $\Sigma A - A_{44}$ |
| Soil and remnants of paramo (5) | $A_{51}$ | $A_{52}$ | $A_{53}$ | $A_{54}$ | $A_{55}$ | $\Sigma A$ | $\Sigma A - A_{55}$ |
| Total year 2 | $\Sigma A$ | $\Sigma A$ | $\Sigma A$ | $\Sigma A$ | $\Sigma A$ | | |
| Gains | $\Sigma A - A_{11}$ | $\Sigma A - A_{22}$ | $\Sigma A - A_{33}$ | $\Sigma A - A_{44}$ | $\Sigma A - A_{55}$ | | |
| **Measure** | **Equation** | | | | | | |
| Parameters for the analysis of land cover changes | Where $A_{xy}$ is the area of change between classes<br>Net change = \|Gains − Losses\|<br>Total change = Gains + Losses<br>Exchange = 2 * MIN (Losses over time 2 − No coverage changes, Gains over time 2 − No coverage changes)<br>Where: | | | | | | |
| Vulnerability | gp = gain/persistence<br>lp = loss/persistence | | | | | | |
| Braimoh Persistence Indices | np= gp − lp<br>Where: gp = gain persistence; lp = loss persistence; np = net persistence change | | | | | | |

*2.4. Projection to 2030*

In order to generate the projection of the study area, the MOLUSCE complement of the QGIS software was applied, which, as its name indicates (Modules for Land Use Change Evaluation), refers to modules for the evaluation of changes in land use. This plugin has been designed to analyze, model, and simulate land use/cover changes effectively, where its graphical user interface has seven main submodules, allowing MOLUSCE to be valued as an intuitive and easy-to-use complement [70].

In order to obtain the cartographic projection for the year 2030 of the coverage present in the study area, the MOLUSCE methodology was applied with four submodules [70]. The first is the input, where the land cover maps (raster) of the last two years of study generated previously with the supervised classification are loaded. The second is the area changes, where the areas of each land cover are calculated, cover change transition matrices, and land cover maps. In the third, the transition potential modeling four modeling methods are available. These are artificial neural networks, weights of evidence, logistic regression and multicriteria evaluation. In the present study, we worked with the first method, where the learning algorithm examines the accuracy achieved in the sample sets (checkpoints) and training validations and also saves the best neural network in memory. The culmination of the training process occurs when the best precision is achieved [71]. The last submodule consisted of the simulation of cellular automata, which generates transition potential maps and the projection map (simulation) of the land cover (output).

A projection to 2030 will allow us to identify a possible scenario if the changes maintain the pattern of the last years studied. Sustainable development objectives were focused on this year, therefore, the associated projections contribute to decision-making to comply with the 2030 Agenda.

## 3. Results

### 3.1. Classification Analysis

The classification of satellite images allowed us to obtain coverage maps for each year of study (Figure 2). In 2000, it was evident that the predominant cover is that which corresponds to soil and the remnants of paramo with an area of 1563.17 ha (33.48%). Subsequently, there were crops, a coverage that has an area of 1186.07 ha (25.40%), followed by pasture cover with 928.39 ha (19.88%) and forest with an area of 875.45 ha (18.75%). The coverage with the smallest area referred to human activity, anthropic coverage, with 116.34 ha (2.49%).

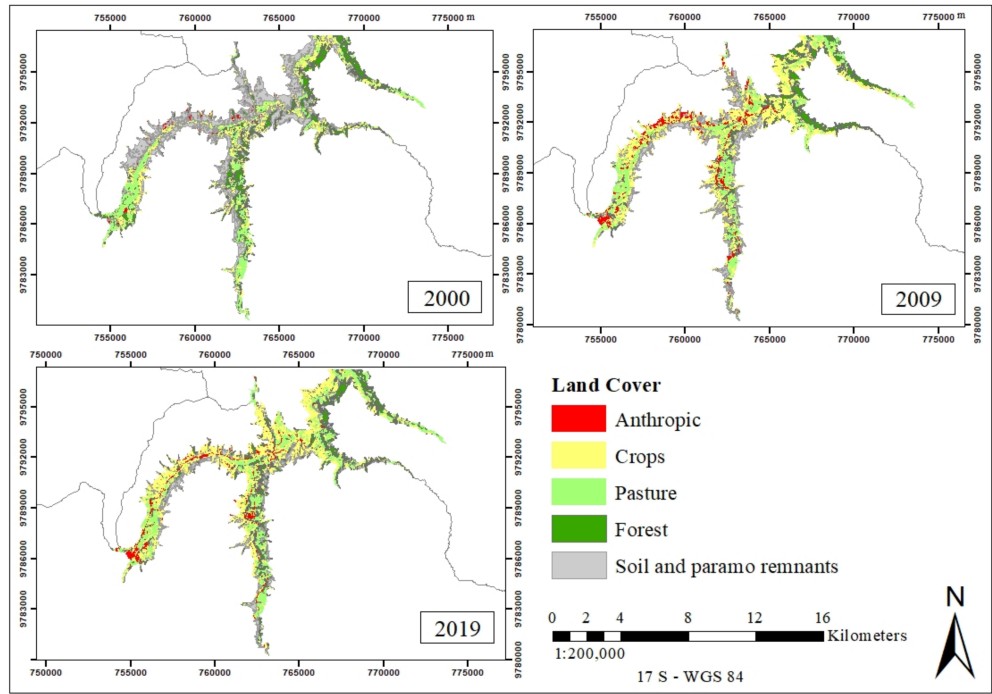

**Figure 2.** Map of the land cover of the study area in 2000, 2009, and 2019.

For 2009, there was evidence of a change in the areas of each type of coverage. There, the predominant coverage is that of crops with 1868.52 ha (40.02), followed by pasture cover with an area of 1072.91 ha (22.98%), forest with 691.49 ha (14.81%), soil and paramo remnants with 713.23 ha. (15.27%), and, with the smallest area, the coverage of the anthropic zone with 323.28 ha (6.92%).

In 2019, the crop cover continued to be the predominant one with an area of 1791.43 ha (38.37%). Similarly, they were followed by pasture cover with 1211.71 ha (25.95%), forest with 813.04 ha (17.41%), soil and paramo remnants with 610.94 ha (13.08%), and the coverage corresponding to the anthropic zones had an area of 242.30 ha (5.19%).

To validate the supervised classification, the precision measures and the kappa index were calculated for each of the classifications in the three years of study. For this, an error matrix was generated for each year. The three coverage classification maps obtained from 2000 and 2009 presented acceptable global accuracy values, since they were in the range of 70–90% for the land cover classifications [72]. In addition, the accuracy of 2019 was 92.86%, considered as almost perfect (Table 2).

On the other hand, there is the value of the kappa index, which qualifies the force of concordance of the study values [73]. In this case, it is attributed to the concordance between the data taken in the field with those obtained from the classification performed by GIS software. The values corresponding to 2000 and 2009, which were 0.68 and 0.80 respectively, determine a considerable agreement of data, while the kappa index for the

year 2019 indicated an almost perfect agreement, being a value of 0.91. Finally, the Table 3 presents the overall accuracy evaluation of this study.

**Table 2.** Accuracy evaluation of the supervised classification.

|  | Pasture | Crop | Anthropic Zone | Forest | Soil and Remnants of Paramo |
|---|---|---|---|---|---|
| **2000** | | | | | |
| % User Accuracy | 87.50 | 29.03 | 94.12 | 80.30 | 79.17 |
| % Producer Accuracy | 64.62 | 50.00 | 100.00 | 86.89 | 76.00 |
| **2009** | | | | | |
| % User Accuracy | 93.55 | 73.53 | 95.65 | 92.11 | 53.33 |
| % Producer Accuracy | 90.63 | 71.43 | 80.00 | 89.74 | 94.12 |
| **2019** | | | | | |
| % User Accuracy | 96.96969697 | 86.21 | 98.11 | 100.00 | 65.22 |
| % Producer Accuracy | 100 | 71.43 | 94.55 | 100.00 | 88.24 |

**Table 3.** Overall accuracy evaluation of the supervised classification and kappa index.

| Year | Precision Measurements | | | Kappa Index |
|---|---|---|---|---|
|  | **Producer Accuracy** | **User Accuracy** | **Overall Accuracy** |  |
| 2000 | 75.50% | 74.02% | 75.24% | 0.68 |
| 2009 | 85.18% | 81.63% | 84.76% | 0.80 |
| 2019 | 90.84% | 89.30% | 92.86% | 0.91 |

### 3.2. Change Trajectory Analysis

With the application of a cross-tabulation matrix, coverage transitions were identified, where the proportions of persistence of each category were located in the cells (blue) of the diagonal of the matrix, and the proportions corresponding to the categories that experienced transitions were found in the remaining cells (Table 4) [68]. Thus, the amount of landscape that did not experience any change in both intervals of time was cultivation, with 3871.6 ha until 2019.

**Table 4.** Cross tabulation matrix with values expressed in percentages (2000–2009).

|  |  |  | 2009 | | | | | | |
|---|---|---|---|---|---|---|---|---|---|
|  |  |  | Pasture | Crop | Anthropic | Forest | Soil and Remnants of Paramo | Total 2010 | Gain |
|  |  | Coding | **10** | **20** | **30** | **40** | **50** |  |  |
| **2000** | Pasture | **1** | 9.79 | 5.14 | 1.16 | 2.26 | 1.54 | 19.88 | 10.09 |
|  | Crop | **2** | 4.89 | 11.11 | 1.56 | 3.22 | 4.62 | 25.40 | 14.29 |
|  | Anthropic | **3** | 0.14 | 1.01 | 0.90 | 0.07 | 0.36 | 2.49 | 1.59 |
|  | Forest | **4** | 4.52 | 5.36 | 0.73 | 6.15 | 1.99 | 18.75 | 12.60 |
|  | Soil and remnants of paramo | **5** | 3.63 | 17.39 | 2.57 | 3.11 | 6.77 | 33.48 | 26.70 |
|  | Total 2017 | | 22.98 | 40.02 | 6.92 | 14.81 | 15.27 | | |
|  | Gain | | 13.19 | 28.91 | 6.02 | 8.66 | 8.50 | | 65.27 |

In the time interval between 2000 and 2009, the transition with the greatest impact corresponded to the change in the soil and remnants from paramo to cultivation, with 17.39% of the total (green cell). These transitions are evident throughout the study area, but in greater quantity in the sections of the rivers located in the Cebadas and Pungalá

parishes. Furthermore, in the period of 2009–2019, the transition with the greatest impact corresponded to that from crop to pasture with 8.06% of the total (green cell) (Table 5). These transitions are illustrated graphically in the cover persistence and transition map. Based on the values resolved in the previous point, it was possible to determine the rate of change in the two time periods of the investigation.

**Table 5.** Cross tabulation matrix with values expressed in percentages (2009–2019).

| | | | 2019 | | | | | | |
| --- | --- | --- | --- | --- | --- | --- | --- | --- | --- |
| | | | Pasture | Crop | Anthropic | Forest | Soil & Remnants of Paramo | Total 2010 | Gain |
| | | Coding | **10** | **20** | **30** | **40** | **50** | Total 2010 | Gain |
| **2009** | Pasture | 1 | 13.63 | 4.60 | 1.00 | 2.83 | 0.92 | 22.98 | 9.35 |
| | Crop | 2 | 8.06 | 21.08 | 1.45 | 4.94 | 4.49 | 40.02 | 18.94 |
| | Anthropic | 3 | 1.02 | 3.05 | 2.03 | 0.43 | 0.40 | 6.92 | 4.89 |
| | Forest | 4 | 1.97 | 3.57 | 0.35 | 6.93 | 1.98 | 14.81 | 7.88 |
| | Soil and remnants of paramo | 5 | 1.27 | 6.07 | 0.36 | 2.28 | 5.29 | 15.27 | 9.98 |
| | Total 2017 | | 25.95 | 38.37 | 5.19 | 17.41 | 13.08 | | |
| | Gain | | 12.32 | 17.29 | 3.16 | 10.48 | 7.79 | | 51.04 |

Table 6 lists the percentage values of the gains, losses, net change, exchange, and total change of the land covers analyzed in the study area. Between the years 2000 and 2009, the following values were identified such as a total change of 65.27%, a net change of 22.14%, and an exchange of 43.13%. The coverage corresponding to cultivation is the one that presented the most interactions with the other coverages, as it had an exchange value of 28.58%. In addition, the categories that registered area losses with respect to net change were forest with 3.94% and the soil and remnants of paramo with 18.20%. Hence, the land covers whose areas presented gains with respect to the net change were pasture with 3.10%, cultivation with 14.62%, and anthropic with 4.43%.

**Table 6.** Rate of change of the time interval of 2000–2009 expressed in percentage terms.

| | Net Change | Exchange | Total Change |
| --- | --- | --- | --- |
| Pasture | 3.10 | 20.18 | 23.28 |
| Crop | 14.62 | 28.58 | 43.20 |
| Anthropic | 4.43 | 3.17 | 7.61 |
| Forest | 3.94 | 17.32 | 21.26 |
| Soil and remnants of paramo | 18.20 | 17.00 | 35.21 |
| Summation | 44.28 | 86.26 | 130.54 |
| Total | 22.14 | 43.13 | 65.27 |

In the period 2009–2019, there was evidence of a total change of 51.04%, a net change of 5.58%, and an exchange of 45.47% of the study area (Table 7). The category that interacted in the highest percentage with the rest, as in the previous period, was that of cultivation, presenting a 34.58% exchange. In addition, the categories that register losses of area with respect to the net change were cultivation with 1.65%, anthropic with 1.73%, and the soil and remnants of paramo with 2.19%. On the other hand, the categories that registered gains in area with respect to the net change were pasture with 2.97% and forest with 2.60%.

In the first period of study (Figure 3), the cover corresponding to the soil and paramo remnants was the one that presented the highest percentage of loss (26.70%) in the first study period due to its transition to the other covers, as listed in Table 8. This value was three times (3.14) greater than the gain percentage (8.50%). In contrast, the class referring to crops was the one that showed the highest percentage of gain (28.91%), which corresponded to twice the percentage of losses (14.29%).

**Table 7.** Rate of change of the time interval 2009–2019 expressed in percentage terms.

|  | Net Change | Exchange | Total Change |
|---|---|---|---|
| Pasture | 2.97 | 18.70 | 21.67 |
| Crop | 1.65 | 34.58 | 36.23 |
| Anthropic | 1.73 | 6.32 | 8.05 |
| Forest | 2.60 | 15.76 | 18.36 |
| Soil and remnants of paramo | 2.19 | 15.58 | 17.77 |
| Summation | 11.15 | 90.94 | 102.09 |
| Total | 5.58 | 45.47 | 51.04 |

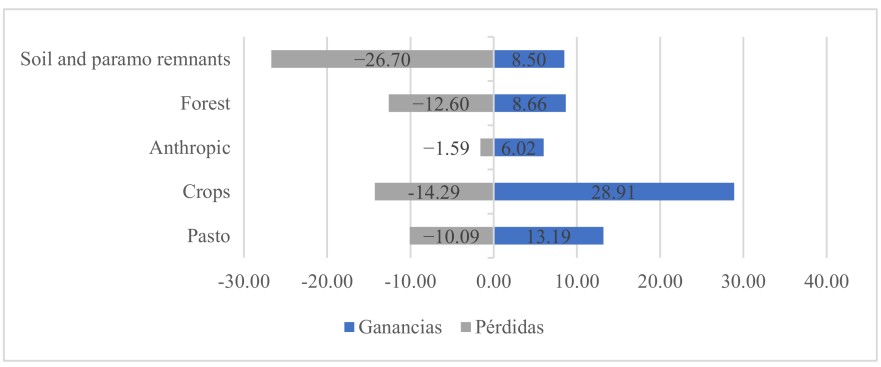

**Figure 3.** Losses and profits in the time interval 2000–2009.

**Table 8.** Coverage transitions in terms of the area in the time interval 2000–2009.

| Code | Change | Area (ha) | Code | Change | Area (ha) |
|---|---|---|---|---|---|
| 11 | Pasture remained | 457.19 | 33 | Anthropic remained | 42.24 |
| 12 | From crop to pasture | 228.23 | 34 | From forest to anthropic | 33.9 |
| 13 | From anthropic to pasture | 6.51 | 35 | From soil and remnants of paramo to anthropic | 119.92 |
| 14 | From forest to pasture | 211.27 | 41 | From pasture to forest | 105.32 |
| 15 | From soil and remnants of paramo to pasture | 169.71 | 42 | From cultivation to forest | 150.58 |
| 21 | From pasture to crop | 239.93 | 43 | From anthropic to forest | 3.33 |
| 22 | Cultivation was maintained | 518.76 | 44 | Forest remained | 287.18 |
| 23 | From anthropic to cultivation | 47.22 | 45 | From soil and remnants of paramo to forest | 145.08 |
| 24 | From forest to farm | 250.39 | 51 | From pasture to soil and remnants of paramo | 71.69 |
| 25 | From soil and paramo remnants to Cultivation | 812.23 | 52 | From cultivation to soil and remnants of paramo | 215.56 |
| 31 | From pasture to anthropic | 54.26 | 53 | From anthropic to soil and remnants of paramo | 17.03 |
| 32 | From cultivation to anthropology | 72.95 | 54 | From forest to soil and remnants of paramo | 92.71 |
|  |  |  | 55 | Soil and remnants of paramo remained | 316.24 |

The forest category presented a loss percentage of 12.60%. As noted in the stratification of the study area, this category included forest plantations in general, and the native forests of the area. Continuing with the descending order of the loss percentages, there was the pasture category, which presented very close percentages of gains and losses at 13.19% and 10.09%, respectively.

In the time interval 2009–2019 (Figure 4), the category with the highest percentage of loss was cultivation with 18.94%. Despite this, this coverage practically lost what it gained. Next, there was the land cover and remnants of paramo, which significantly decreased its percentages of transitions compared to the period analyzed above. However, the tendency to present a higher percentage of loss (9.98%) than that of gain (7.79%) was maintained.

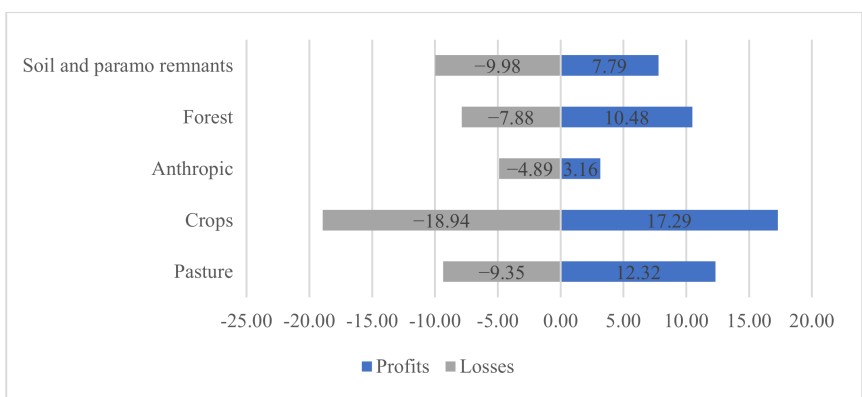

**Figure 4.** Losses and profits in the time interval 2009–2019.

Regarding pasture cover, a higher percentage of gains (12.32%) than losses (9.35%) was observed, maintaining a trend similar to the previous period. In the forest category, in this period of time, it presented a higher percentage of the gains (10.48%) than the losses. The same occurred with anthropic coverage, quite the opposite of the previous period, where its percentage of losses (4.89%) was slightly higher than the percentage of gains (3.16%). The corresponding areas are presented in the Tables 8 and 9.

**Table 9.** Coverage transitions in terms of area in the time interval 2009–2019.

| Code | Change | Area (ha) | Code | Change | Area (ha) |
|---|---|---|---|---|---|
| 11 | Pasture remained | 636.34 | 33 | Anthropic remained | 94.76 |
| 12 | From crop to pasture | 376.49 | 34 | From forest to anthropic | 16.43 |
| 13 | From anthropic to pasture | 47.52 | 35 | From soil and remnants of paramo to anthropic | 16.77 |
| 14 | From forest to pasture | 92.07 | 41 | From pasture to forest | 132.37 |
| 15 | From soil and remnants of paramo to pasture | 59.28 | 42 | From cultivation to forest | 230.44 |
| 21 | From pasture to crop | 214.65 | 43 | From anthropic to forest | 20.03 |
| 22 | Cultivation was maintained | 984.18 | 44 | Forest remained | 323.56 |
| 23 | From anthropic to cultivation | 142.38 | 45 | From soil and remnants of paramo to forest | 106.65 |
| 24 | From forest to farm | 166.85 | 51 | From pasture to soil and remnants of paramo | 42.82 |
| 25 | From soil and paramo remnants to cultivation | 283.38 | 52 | From cultivation to soil and remnants of paramo | 209.81 |
| 31 | From pasture to anthropic | 46.73 | 53 | From anthropic to soil and remnants of paramo | 18.59 |
| 32 | From cultivation to anthropology | 67.61 | 54 | From forest to soil and remnants of paramo | 92.57 |
| | | | 55 | Soil and remnants of paramo remained | 247.15 |

Based on the coverage change analysis performed, the tables and maps that indicate the interactions recorded in this study are presented below (Figures 5 and 6).

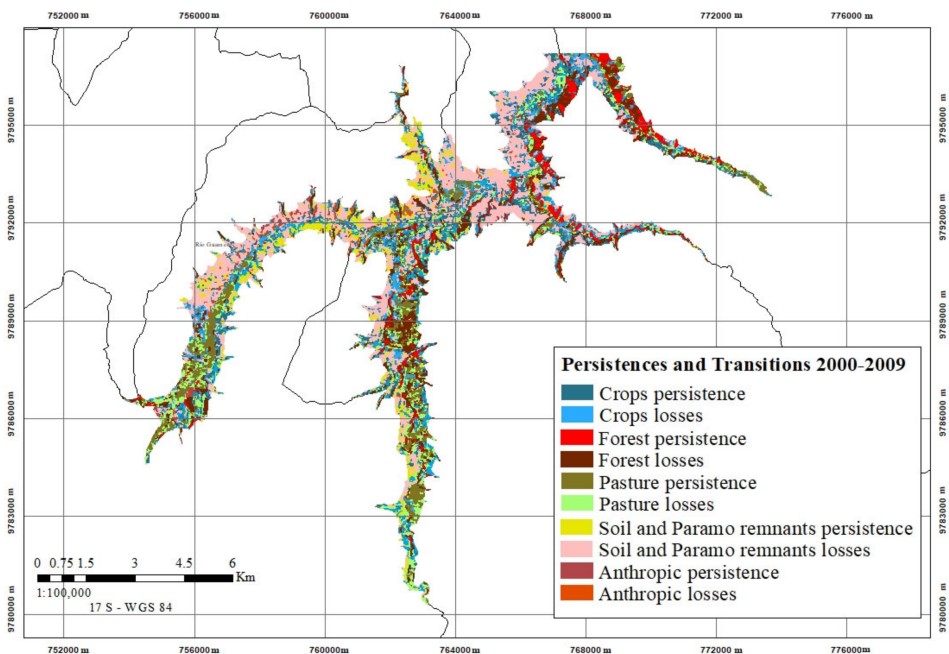

**Figure 5.** Map of coverage persistence and transitions between 2000 and 2009.

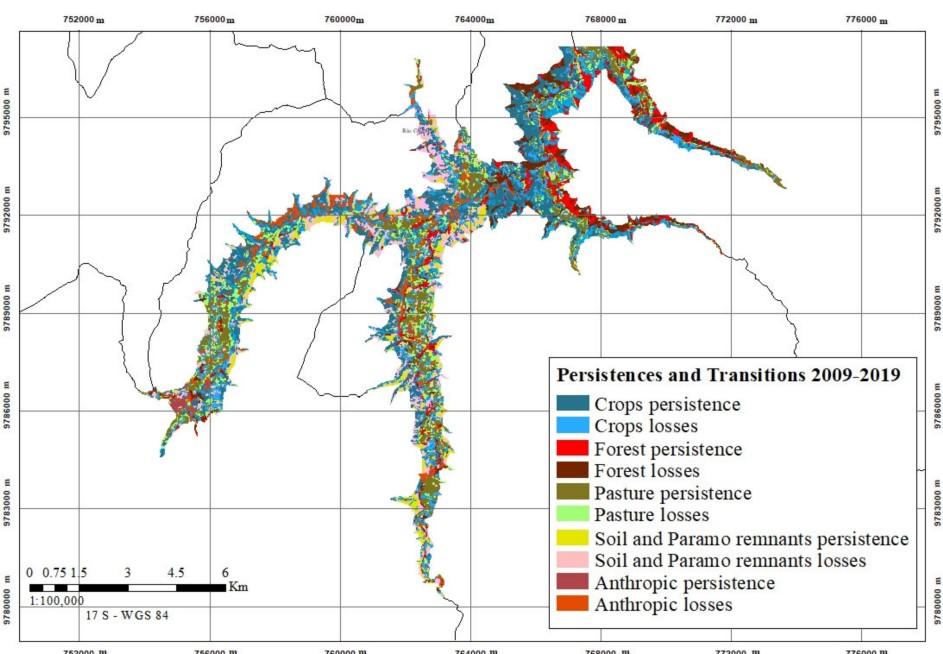

**Figure 6.** Map of coverage persistence and transitions between 2009 and 2019.

The persistence of the losses and gains were evaluated according to the tendency that the proportions in each category presented to decrease or increase. Table 10 lists the persistence coefficients of losses, gains, and net changes [69].

**Table 10.** Persistence coefficients of the gains, losses, and net changes in the time intervals 2000–2009 and 2009–2019.

|  | 2000–2009 | | | 2009–2019 | | |
|---|---|---|---|---|---|---|
|  | gp | lp | np | gp | lp | np |
| Pasture | 1.35 | 1.03 | 0.32 | 0.90 | 0.69 | 0.22 |
| Crop | 2.60 | 1.29 | 1.32 | 0.82 | 0.90 | −0.08 |
| Anthropic | 6.65 | 1.75 | 4.90 | 1.56 | 2.41 | −0.85 |
| Forest | 1.41 | 2.05 | −0.64 | 1.51 | 1.14 | 0.38 |
| Soil and remnants of paramo | 1.26 | 3.94 | −2.69 | 1.47 | 1.89 | −0.41 |

gp = gain persistence, lp = loss persistence, np = net persistence change.

When the values of gp and lp are greater than 1, they indicate that there is a higher tendency to increase or decrease than to persistence [69]. In this way, in the period 2000–2009, all categories had a greater tendency to win than to persist, with the anthropic category being the one with the greatest impact (6.65). Similarly, this occurred with the tendency to lose, however, in this case, the most important value was the one belonging to the category soil and paramo remnants.

In the 2009–2019 period, except for the pasture and crop categories, all categories tended to win and lose more than to persist. Hereby, the coefficient corresponding to the anthropic class in the first period was a very high value compared to the others, even when compared to the value of the same class in the following period. Additionally, in those years, there was also a persistence of a loss of soil and paramo remnants.

*3.3. Cartographic Projection for the Year 2030*

The projection for the year 2030 in the study area was obtained using the artificial neural network modeling method of the MOLUSCE complement. Where, in the first place, a neighborhood of nine pixels, a learning rate of 0.1, and a maximum of 1000 iterations were established [43], in order to obtain the neural network learning curve that was found, as illustrated in Figure 7, where the green color represents the training dataset and the red lines are from the validation dataset.

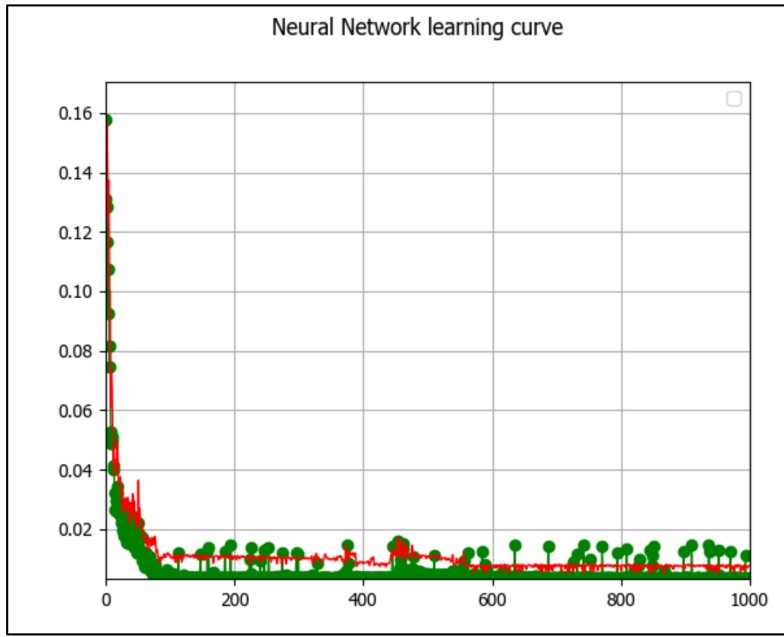

**Figure 7.** Neural network learning curve.

The general minimum validation error was 0.00527 and the current validation kappa was 97.33% for the simulation map of the land cover of the year 2030 (Figure 8).

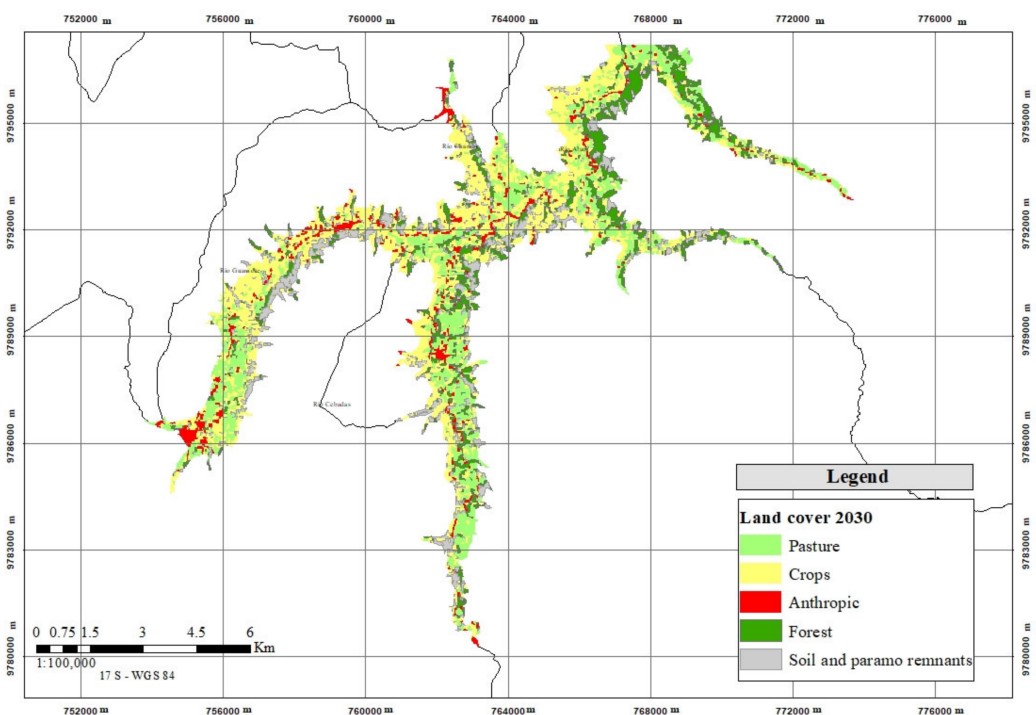

**Figure 8.** Map of the land cover projection for the year 2030.

Based on the simulation carried out, it is predicted that for the year 2030, the predominant cover will continue to be that of cultivation with an area of 1770.52 ha (37.92%), followed by pasture cover with 1191.72 ha (25.52%), forest with 757.53 ha (16.22%), and finally the soil and paramo remnants with 647.88 ha (13.87%) and the anthropic area with 301.77 ha (6.46%).

Table 11 presents a compilation of the values that determine the variation in the area of each of the categories in the period 2000–2019, based on previously obtained information. Additionally, the values belonging to the projection of the study area in the future (2030) are presented. Thus, based on the simulation carried out, it is predicted that between 2019 and 2030, there will be the following variations in each of the studied coverages (Figure 9):

**Table 11.** Land cover change between the years 2000 and 2019 and its prediction to 2030.

| Year | 2000 | | 2009 | | Exchange rate 2000–2009 | | 2019 | | Exchange rate 2009–2019 | | 2030 | | Exchange rate 2019–2030 | |
|---|---|---|---|---|---|---|---|---|---|---|---|---|---|---|
| Coverage | Area (ha) | % | Area (ha) | % | Area (ha) | % | Area (ha) | % | Area (ha) | % | Area (ha) | % | Area (ha) | % |
| Pasture | 928.39 | 19.88 | 1072.91 | 22.98 | 144.52 | 3.10 | 1211.71 | 25.95 | 138.80 | 2.97 | 1191.72 | 25.52 | −19.99 | −0.43 |
| Crop | 1186.07 | 25.40 | 1868.52 | 40.02 | 682.45 | 14.62 | 1791.43 | 38.37 | −77.08 | −1.65 | 1770.52 | 37.92 | −20.91 | −0.45 |
| Anthropic | 116.34 | 2.49 | 323.28 | 6.92 | 206.94 | 4.43 | 242.30 | 5.19 | −80.98 | −1.73 | 301.77 | 6.46 | 59.47 | 1.27 |
| Forest | 875.45 | 18.75 | 691.49 | 14.81 | −183.96 | −3.94 | 813.04 | 17.41 | 121.56 | 2.60 | 757.53 | 16.22 | −55.51 | −1.19 |
| Soil and remnants of paramo | 1563.17 | 33.48 | 713.23 | 15.27 | −849.95 | −18.20 | 610.94 | 13.08 | −102.29 | −2.19 | 647.88 | 13.87 | 36.94 | 0.79 |

The category that presents the highest percentage of variation will be anthropic, with a gain of 1.27%, followed by forest with an area loss of 1.19%, the soil and paramo remnants will present a gain of 0.79%, the crop cover will present a loss of 0.45%, and finally, the pasture cover will also present a decrease of 0.43% in its area.

Table 12 presents the prediction, for the year 2030, of the areas corresponding to the transitions and persistence of land cover.

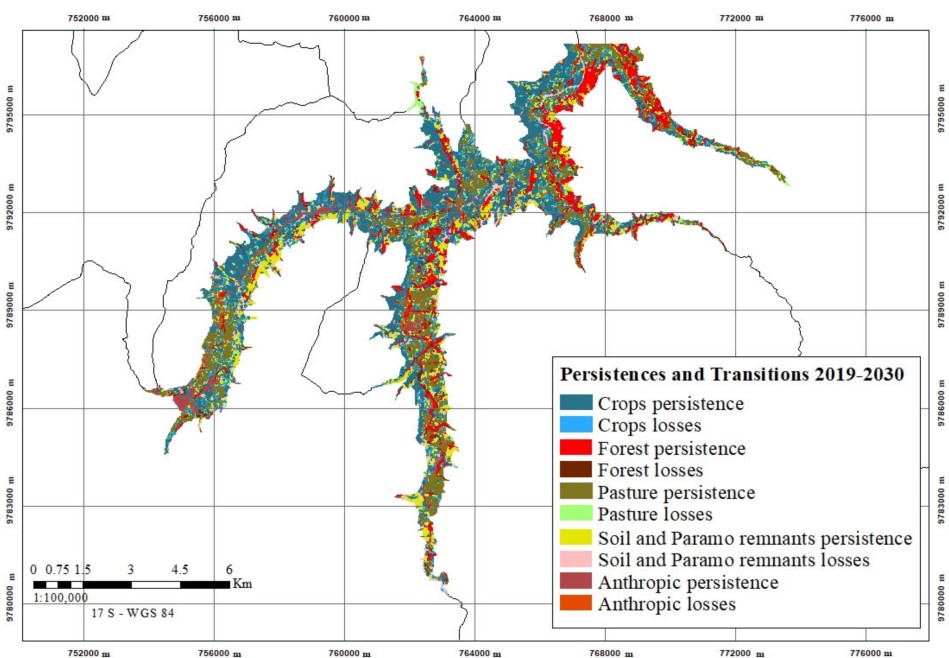

**Figure 9.** Map of coverage persistence and transitions between 2019 and 2030.

**Table 12.** Coverage transitions in terms of area in the time interval of 2019–2030.

| Code | Change | Area (ha) | Code | Change | Area (ha) |
|---|---|---|---|---|---|
| 11 | Pasture remained | 919.28 | 33 | Anthropic remained | 145.64 |
| 12 | From crop to pasture | 161.03 | 34 | From forest to anthropic | 20.01 |
| 13 | From anthropic to pasture | 27.36 | 35 | From soil and remnants of paramo to anthropic | 23.14 |
| 14 | From forest to pasture | 61.97 | 41 | From pasture to forest | 67.72 |
| 15 | From soil and remnants of paramo to pasture | 22.08 | 42 | From cultivation to forest | 63.45 |
| 21 | From pasture to crop | 160.31 | 43 | From anthropic to forest | 11.91 |
| 22 | Cultivation was maintained | 1366.93 | 44 | Forest remained | 571.32 |
| 23 | From anthropic to cultivation | 54.25 | 45 | From soil and remnants of paramo to forest | 43.13 |
| 24 | From forest to farm | 63.97 | 51 | From pasture to soil and remnants of paramo | 20.19 |
| 25 | From soil and paramo remnants to cultivation | 125.06 | 52 | From cultivation to soil and remnants of paramo | 123.81 |
| 31 | From pasture to anthropic | 51.96 | 53 | From anthropic to soil and remnants of paramo | 11.09 |
| 32 | From cultivation to anthropic | 61.02 | 54 | From forest to soil and remnants of paramo | 86.16 |
| | | | 55 | Soil and remnants of paramo remained | 406.63 |

## 4. Discussion

Landscape changes have been detected with greater emphasis from the 21st century, where each sector can present a different scenario. However, the increase in population, and therefore economic expansion, are the main factors of change in surface coverage [74], which have also been detected in the present study.

In the land cover classification of the time interval 2000–2009, it was detected that soil and paramo remnants had the highest area of loss, which can be attributable to the paramo degradation due to the fact that a large part of the paramos in the province of Chimborazo has been exposed to pressure through common practices of agriculture, livestock, and reforestation with introduced species. Furthermore, the burning of most of the straw is also a common practice, whose purpose is the regrowth of tender straw to feed cattle, thus increasing its productivity [75]. Despite, in the second period of study 2009–2019, the transitions having decreased, the pattern of the first period was maintained. Furthermore,

this is mainly supported by the degradation of the paramos, which is linked to the gains of the other categories, because in addition to replacing the paramos, they also occupy the land without coverage (soil).

On the other hand, crops presented the highest gain of area. For this reason, it is appropriate to cite the Law of Agrarian Reform and Colonization, which entered into force in Ecuador in July 1964, proposing a new concept of obtaining land use rights. Therefore, up to 30 years after its application, the Ecuadorian Andes underwent a significant transformation, giving rise to the expansion of agricultural activities [76]. Nonetheless, regarding the loss of this cover, adding to the activities above-mentioned, the ash from volcanic eruptions has also been a problem that has contributed to the loss of crops [36,37].

In the case of the percentage of forest loss, the causes were also related to the causes of the categories described in the previous paragraphs, considering that the representative native forests of the study area were those located in the paramo zones. In the same way, colonization and agricultural expansion have caused the replacement of these forest covers. The profit value (8.66%) was supported by reforestation and afforestation activities, applied with different objectives such as the "Afforestation and reforestation project of the Chambo River sub-basin", which sought to contribute to the environmental conditions in the sub-basin, using native species for protection and exotic for production. This completed project presented its results in 2008 [77].

The close percentages of gains and losses of pasture cannot represent a significant impact in the area in both periods of study. However, the floods in the sub-basin, especially in the middle part and in the part of the mouth, are presented as an important threat for this category, as it has been determined that along the Chambo River sub-basin, 28% of the areas affected by flooding correspond to planted pastures [78]. The greater gain than loss can be attributed to the afforestation and reforestation activities in previous years.

Regarding anthropic coverage, in the first interval of time, this class had a great tendency to win due to the typical population increase of societies. As the years went by, in the second period, their tendency to lose was greater than their tendency to win, a situation supported by the migration of the rural population to urban towns [36], considering that most of the study area corresponded to rural areas. Nevertheless, its percentage of loss was the one with the least impact compared to the other categories. Furthermore, according to the exchange rates calculated in the study periods, it is possible to determine that the future projection indicates the lowest values of coverage losses compared to previous years.

Comparing the results obtained, based on the measures of accuracy of the supervised classification, with Enderle and Weih [79], who developed a land cover classification of watersheds in Arkansas (USA) using supervised and unsupervised classification with images Landsat 7 (1995), in the present study, global accuracies between 75 and 93% were obtained, where the lowest accuracy being that corresponding to the Landsat 7 image of the year 2000, whose value was very close to that obtained in the supervised classification of the study in comparison. The same authors obtained a considerably lower global accuracy value of 40.94% in the unsupervised classification of the same image.

Similarly, Mohammady et al. [80] compared unsupervised and supervised land cover classification of a Baghsalian watershed in Iran with Landsat 7 images (2010), where their global accuracy values were between 58 and 78% in the first case and between 81 and 84% in the second case. In addition, the kappa index ranged between 0.57 and 0.74 and 0.79 and 0.82 as appropriate. Despite the fact that the supervised classification method was different to that in the present study, it was possible to determine that the supervised classification presented an appreciable advantage over the unsupervised classification of Landsat images, a situation repeated in other similar works such as in the case of Nijhawan et al. [81], who used the same method of the present investigation, and whose precision value was greater than that of the other types of classification.

The changes in coverage detected in this research are comparable with others performed in Ecuador such as in the Southern Andes [82], where their results agreed that the expansion of cultivable borders predominated over the rest of the coverage, in addition,

the anthropic activity corresponding to the construction of roads and buildings was one of the most relevant causes of deforestation, without omitting its replacement by agricultural crops and pasture. According to Viña et al. [83] in their satellite evaluation of deforestation rates, carried out on the Colombia–Ecuador border between the years 1973 and 1996, they attributed the cause of deforestation to human colonization, a situation that is consistent with the results of this study of the first period (2000–2009), where the profit values of the anthropic zone were really high when compared with the rest of the coverages, and even in the second study period. Additionally, in an evaluation of changes in land use in the entire Chambo River basin between 1979 and 2014 [84], it was determined that one of the main transitions of coverage in that period was that of paramo to agricultural crops, a statement that agrees with the first study period in this work, where this transition was the one with the greatest area of change.

A further point is the projection made for the year 2030, where it would be expected that the area increase value of the anthropic zone would rise with respect to the first study period, where its value was representative. However, despite the fact that the projection estimated that there will be an increase, its value was not significant. At this point, we detected an uncertainty level in this MOLUSCE projection, because with the input of other variables such as a population growth rate between the years studied, the projected values can get even closer to a more accurate projection. This was corroborated by the study by Cosuegra et al. [85], which determined that migration from rural to urban areas in the different provinces of Ecuador has increased significantly since the 1970s due to the process of the development and growth of the main cities.

To complement this work, it would be important to perform a complementary study that relates spatial patterns with the water quality of the channels of the upper middle basin of the Chambo River, in order to link it with the impacts generated by changes in land cover. Similarly, these studies should involve an analysis of the effect of climate variability on the land use change and vice versa, and also on the livelihoods in the study area. This is to evaluate its impact on the change in coverage and determine the behavior of the population in the face of changes in coverage that have occurred over the years.

## 5. Conclusions

- Through a supervised classification using the maximum likelihood algorithm, the study area located within the upper middle basin of the Chambo River sub-basin corresponding to its source area was stratified, where five predominant land covers were identified, being pasture, crops, soil-remnants of paramo, forest, and anthropic.
- The obtained maps allowed us to identify that in the first period (2000–2009), the soil cover-paramo remnants presented the highest percentage of loss and the crop cover the highest percentage of gain, and in the second period (2009–2019), the crop presented the highest percentages of gains and losses, transitions were attributed to population growth, afforestation, reforestation, deforestation and agricultural activities, volcanic eruptions, land colonization, and the expansion of agricultural activity.
- The cartographic projection of the coverage in the riparian ecosystem for the year 2030 allowed us to determine that the anthropic zone and soil-remnants of the paramo will increase their area and the areas of pasture, cultivation, and forest will be reduced as these percentages were insignificant and those with the least impact compared to the periods corresponding to previous years.

**Author Contributions:** Conceptualization, J.E.-P. and F.C.; methodology, J.E.-P. and F.C.; software, J.E.-P. and F.C.; validation, J.E.-P., F.C. and M.E.; formal analysis, J.E.-P.; investigation, J.E.-P. and T.T.; resources, M.E.; data curation, J.E.-P.; writing—original draft preparation, J.E.-P.; writing—review and editing, T.T.; visualization, J.E.-P. and T.T.; supervision, M.E.; project administration, M.E.; funding acquisition, J.E.-P., F.C., M.E. and T.T.; All authors have read and agreed to the published version of the manuscript.

**Funding:** This research received no external funding.

**Conflicts of Interest:** The authors declare no conflict of interest.

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
