# Peer review of "Spatial Dynamics of the Shore Coverage within the Zone of Influence of the Chambo River, Central Ecuador"

_land, doi:10.3390/land12010180_

Round 1

Reviewer 1 Report

Overall this paper is good as it provide an update and interesting take regarding the land use study using GIS and remote sensing. However, it would be good to have a residual map in which you can see the differences of land use between the two data (e.g. 2009 and 2019) very clearly in the map. This is just my suggestion though.

Author Response

Reviewer #1

Overall this paper is good as it provide an update and interesting take regarding the land use study using GIS and remote sensing. However, it would be good to have a residual map in which you can see the differences of land use between the two data (e.g. 2009 and 2019) very clearly in the map. This is just my suggestion though.

Resp.: Dear expert reviewer, thanks a lot for your kind words. We have speculated to rather reduce than extend the amount of maps we have in the manuscript. In fact your colleague reviewers strongly recommended to compile the maps of different times, as we did. Sorry.

Reviewer 2 Report

This research aims to evaluate land-cover change between 2000 and 2019 in riparian zones of the Chambo river in Ecuador and to project trends to 2030. While to my knowledge understanding land-cover dynamics in and around Andean river remains a topic poorly explored in the scientific literature, the manuscript in its current stage presents significant errors that must be revised before considering this work for publication. See details below my comments for each section:

Introduction

General comments:

In general, the introduction fails to provide the current stage of knowledge around the research question of this study and its broader context. Why is it important to understand land-cover change in Andean rivers? What’s the current stage of knowledge of land-cover change of riparian zones in these ecosystems? What are the drivers of these changes? Why is understanding land-cover change in riparian zones important? What are the gaps of knowledge?  These are some of the questions I think the introduction should address given the research question of this study (i.e. “evaluate the spatial dynamics of the riparian coverage of the area of influence of the Chambo river”). I encourage the authors to add critical information to better contextualize and provide background for this research, including: human-impacts in riparian zones broadly and specifically in the Andes (or mountainous ecosystems if information about the Andes is limited), land-cover change and its drivers in Andean riparian zones. For instance, many of the capitals of the Andean countries are in the Andes, which has resulted in drastic transformation of these rivers. These areas are also places rich in fertile soils where agriculture has been practiced for centuries, etc.

Specific comments:

-   Page 1, line 34: What do you mean by complex? Complex in which way, ecologically, socially. Clarify.

-   Page 2, lines 47-48:The relationship between land-cover change and climate is not clearly explained.

-   Page 2, lines 49-54: Is this information related to the study area, the region, the globe? The links between land-cover change and climate change are not clearly provided.

-   Page 2, lines 61-66: No need to justify the use of remote sensing.

-   Page 2, lines 68-77: No need to explain the location in Ecuador and major geographic regions of Ecuador here, move to the Study Area section. Same with information about volcanic activity.

-   Page 2, lines 78-84: No need to report on land-cover changes studies done at the country level, just state that natural areas around the Chambo river have been lost between years X and Y and support that claim with the literature.

-   Page 2, lines 89-93: The relationship between climate change and land-cover change is not clearly explained and it might not be totally relevant to this study.

-   Page 2-3, lines 100-102: No need to mention which software or systems of analysis used for data analysis.

Some language comments:

-   Rough transitions that could use better phrases or words (Herby… Based on the aforementioned”)

-   Excessive use  of run-on sentences that do are hard to follow (e.g. line 89-92)

Study Area

Overall comments:

-   Page 3, lines 105-114: Condense information. Focus on describing the location of the river.

-   Page 3, lines 117-128: Condense information. Focus on detailing how a digital elevation model was used to identify a “water network”. Provide details about the resolution of the digital elevation model and the source used to obtain it. There is no need to be so specific about the functions used within a GIS software (e.g. Map Algebra, Strem [Stream?] Order tool, etc.) Rather, describe the analytical process undertaken and end the section with the phrase  “Analyses were conducted in NAME OF SOFTWARE”

-   Page 4, lines 129-137: Merge this information with that of the paragraph of Page 3, lines 117-128. Moreover, for consistency, I suggest the use of DEM data to delimit the study area based on altitude, not the use of contour lines.

-   Cite Figure 1 at the end of Study Area.

-   Figure 1: Add Chimborazo province name to the area highlighted in green in the inset maps. Increase fonts.

Pre-processing of satellite images (Rename to “Satellite Image Pre-processing)

Overall comments:

-   Page 4, lines 139-146: Rephrase. This is not an “additional” analysis. Image Pre-processing is a critical step. The authors need to justify why using three different sensors is justified. There is no need to provide information about projections.

-   Table 1. Information that is constant about each sensor is not necessary. The only critical part of this table is the spatial resolution of the data. Therefore, I suggest deleting this table and incorporating only information about the spatial resolution in the text.

-   Page 4 line 148 – Page 5 line 166: There is no need to provide equations about Landsat image processing with this level of detail, just state that radiance values were converted to top of atmospheric reflectance using the dark subtraction approach in ENVI

-   Page 5, lines 167-176: If the authors had access to ENVI, why were classification analyses done in QGIS? ENVI is a more powerful software for the analysis of spectral data. Also, it is not clear whether all the bands in the sensors were used for classification or just the green, blue and red bands. I don’t see any valid reason to use just a subset of bands in a supervised classification. The authors will have to provide a strong justification if that is the case. Finally, before describing any classification process, the authors need to explain how the training data were collected, how many classes were used and why.

-   Page 5, lines 177-187: Merge this information with that of the previous paragraph. Just indicate that maximum likelihood was used and justify its use. There are more powerful classification approaches nowadays (e.g. machine learning algorithms), so the authors need to justify the use of more conventional approaches.

-   The information between Page 5, lines 167 and Page 5, lines 187 does not correspond to image pre-processing analysis. It describes classification analyses.

Section 23 (I assume 2.3)

-   Rename to something more meaningful (e.g. Classification Analyses)

-   Page 5, line 188 – Page 6, line 209 and Table 2: It is not necessary to provide this level of details about accuracy analysis. Delete this table, describe the classification approach used and then detail how model accuracy was assessed. Just mention and briefly define user, producer, overall accuracy and Kappa.

-   Table 3 is unclear and should be re-done. Do not use generic terms (e.g. Class 1, Class 2, etc.). Instead, use the land-cover classes used in classification analysis (e.g. crops, forest, etc.)

-   Page 6, line 209 – Page 7, line 228 and Table 3: I disagree with the change trajectory approach used. Instead of using cross-tabulations for every pair of years, my suggestion would be to make a change trajectory analysis between the three years. With 5 classes and three points in time, there are 125 possible transitions, which is a huge number. Therefore, I suggest focusing on a couple of meaningful transitions, those critical to explaining changes in riparian zones. For example, “Forest Losses” could encompass all transitions of a “Forest” class over the three time periods. Similar with other land-cover types, with emphasis on those important to understand critical processes of land-cover change in riparian vegetation. Label these transitions something meaningful, e.g. crop gains, etc., and report these results in terms of area and percentages.

-   Until this point in the paper there has not been mentioned which land-cover classes are being studied. Please add that information when detailing the classification approach. Also, include information about the training data: how was it collected and how many points per class per used.

Section 2.4 (rename)

-   I am not familiar with the MOLUSCE approach, but I think it is important to explain the level of uncertainty associated with making a projection when only three points in time are available (or maybe just two based on the description of the approach). Moreover, why 2030? Please justify. Explain inputs, outputs and level of uncertainty of the model.

Section 3.1 (Rename something more meaningful related to classification analysis)

-   Page 7, line 251 – Page 8, line 287: This information belongs to the Methods section. Condense and move the Methods section. See comments above.

-   Page 7, line 288 – 299: Condense. There is no need to repeat the imagery used. Start by describing the results of the classification analyses in terms of areas. This information needs to be accompanied by the accuracy of each classification.

-   Merge Figures 2, 3 and 4 into a single composite image with three maps (one for each classification result) but a single legend and scale. Add a north arrow and increase the font.

-   Page 10, line 311-318: Information about accuracy assessment should accompany the description of the classification results.

-   Table 4: This is a critical issue. It is not enough to provide overall classification accuracy and Kappa. The authors need to report classification accuracies and Kappa for each class for each classification. Ideally, the confusion matrix of the training data should be added too if space and word limit allows. I am particularly curious about the accuracy values of paramo, crops and pasture, which are critical for this study.

-   Page 10, line 321-327 and Table 5: In general, delete. Accuracy information should be provided when classification results are being reported. There is no need to describe what different ranges of Kappa mean, just state “how good” these values are considered when they are mentioned in classification results.

-   Page 10, line 330 – Page 17, line 455: Make a new section for the change trajectory analyses. I am going to provide less detailed comments about this part because I disagree with the change trajectory approach used.

The authors should start this section by indicating the amount of the landscape that did not experience any changes over the time frame of the study. Then, they could focus on critical transitions that help to understand the dynamics of land-cover types in the study area and their implications (for e.g. for biodiversity, conservation, water protection, etc.). As it is, it is difficult to understand what are the main changes in land-cover over the 2000-2009-2019.

Review terminology, e.g. profits. “Gains” might be a better word.

The information presented in Tables 8 and Table 9 is the same as that presented in Figures five and six. This is redundant. I suggest keeping the Figures and deleting the tables (but make sure the labels of both axis are properly labeled)

Do not mention locations that have not been previously introduced (e.g. line 337)

I suggest presenting details of important transitions in tables and only critical transitions in the figures. As they are, Figure 7 and 8 are very difficult to interpret.

-   I strongly disagree with having Results and Discussion merged. I suggest keeping the results section to describe research outcomes and add a Discussion section to contextualize and explain findings within a bigger body of literature. For instance, the author should discuss the use of MOLUSCE. Where else has this technique been used? For what purposes? What short of uncertainties do we have of the projected values? Why 2030? In regards to land-cover change, what are the main drivers of these changes? How do land-cover change patterns compare with other similar studies?

-   Page 20, lines 542-556: The conclusions should reflect key findings related to the research question being investigated: “evaluate the spatial dynamics of the riparian coverage of the area of influence of the Chambo river”. Do not mention Kappa values or other metrics, concentrate on key findings. What are the main processes of land-cover change and their implications?

Abstract

-   Describe what the MOLUSCE method does or if possible avoid mentioning it. A reader should be able to understand what was done without having to search what the MOLUSCE technique is.

-   Based on all comments provided, clearly describe land-cover classification results and associated accuracies per class, key land-cover changes patterns between 2000-2009-2019 and how these findings fit within a bigger body of literature.

-   Improve the selection of keywords. For instance, “remote sensing” is very generic and does not provide. Think about specific terms that would be used to search for paper related to this specific topic.   

Author Response

Reviewer#2

This research aims to evaluate land-cover change between 2000 and 2019 in riparian zones of the Chambo river in Ecuador and to project trends to 2030. While to my knowledge understanding land-cover dynamics in and around Andean river remains a topic poorly explored in the scientific literature, the manuscript in its current stage presents significant errors that must be revised before considering this work for publication. See details below my comments for each section:

Resp.: Dear expert reviewer, thank you for your introductory words about our manuscript and the topic we deal with. This demonstrates that you do know the importance of the region and the issues we dealt with. Even more so, we have almost never see a such detailed and extensive review before, which made us proud that someone took the time to show us a better way to express our findings as you will see, with our detailed responses in each one of those remarks you have left us, allowing us to improve greatly our manuscript.

Introduction

General comments:

In general, the introduction fails to provide the current stage of knowledge around the research question of this study and its broader context. Why is it important to understand land-cover change in Andean rivers?

Resp.: Andean rivers have due to the past and current hydro-meteorological activity as well as geodynamic movements (and corresponding setting), plus the human involvement, some obvious changes to indicate. A study like ours may bring some light to areas usually forgotten or ignored, which may be used for comparison to other actively evolving and or changing regions in the vicinity (South America) as well as worldwide.

What’s the current stage of knowledge of land-cover change of riparian zones in these ecosystems?

Resp.: (66-65) In these lines we explained the identified land-cover changes. Even, it is indicated with the use of spatial dynamics in such kind of studies, which may be applied in technologies or plans in order to reduce wrong or inadequate land-cover change activities.

What are the drivers of these changes?

Resp.: See about this Lines 43,44,45.

Why is understanding land-cover change in riparian zones important?

Resp.: The changes in land cover riparian zones with the subsequent loss of native plant cover cause the fragmentation of habitats and ecosystems [8, 26], which negatively affects ecological processes and endangers the species of flora and fauna associated with the place. (lines 59-62)

What are the gaps of knowledge? 

Resp.: Unfortunately, there is a lack of sensors dedicated to these specific areas, therefore the images do not present the adequate resolutions as we would wish for.

These are some of the questions I think the introduction should address given the research question of this study (i.e. “evaluate the spatial dynamics of the riparian coverage of the area of influence of the Chambo river”). I encourage the authors to add critical information to better contextualize and provide background for this research, including: human-impacts in riparian zones broadly and specifically in the Andes (or mountainous ecosystems if information about the Andes is limited), land-cover change and its drivers in Andean riparian zones. For instance, many of the capitals of the Andean countries are in the Andes, which has resulted in drastic transformation of these rivers. These areas are also places rich in fertile soils where agriculture has been practiced for centuries, etc.

 Resp.: Plenty of information has been added extensively, as you will certainly see throughout the introduction.

Specific comments:

-    Page 1, line 34: What do you mean by complex? Complex in which way, ecologically, socially. Clarify.

Resp.: These are specifically ecologically complex systems, as they constitute transition zones or interfaces, as it is indicated by an ecotone located between terrestrial and aquatic zones.

-    Page 2, lines 47-48:The relationship between land-cover change and climate is not clearly explained.

Resp: Line 47-51. Thus, land use, an activity characterized by the type of cover, and particularly plant cover, play an important role in regulating the climate and different phases of the hydrological cycle, because the surface characteristics of ecosystems regulate the radiation balance and the mass and energy flows between the soil and the atmosphere, the intensity of the wind with its roughness and the humidity of the surface layer

-    Page 2, lines 49-54: Is this information related to the study area, the region, the globe? The links between land-cover change and climate change are not clearly provided.

Resp: It is related to the region which can be associated with the study area, also the references are from other countries’ studies where we found the similarities. Furthermore, the links between land-cover change and climate change are now explained again, in lines 47-51.

-    Page 2, lines 61-66: No need to justify the use of remote sensing.

Resp: OK

-    Page 2, lines 68-77: No need to explain the location in Ecuador and major geographic regions of Ecuador here, move to the Study Area section. Same with information about volcanic activity.

Resp: OK

-    Page 2, lines 78-84: No need to report on land-cover changes studies done at the country level, just state that natural areas around the Chambo river have been lost between years X and Y and support that claim with the literature.

Resp: OK (64-66)

-    Page 2, lines 89-93: The relationship between climate change and land-cover change is not clearly explained and it might not be totally relevant to this study.

Resp: OK. Lines have been deleted.

-    Page 2-3, lines 100-102: No need to mention which software or systems of analysis used for data analysis.

Resp: Ok. Mention has been eliminated.

Some language comments:

-    Rough transitions that could use better phrases or words (Herby… Based on the aforementioned”)

-    Excessive use  of run-on sentences that do are hard to follow (e.g. line 89-92)

Resp.: Thank you, has been modified as requested.

Study Area

Overall comments:

-    Page 3, lines 105-114: Condense information. Focus on describing the location of the river.

Resp.: OK as demonstrated in lines 90-93

-    Page 3, lines 117-128: Condense information. Focus on detailing how a digital elevation model was used to identify a “water network”. Provide details about the resolution of the digital elevation model and the source used to obtain it. There is no need to be so specific about the functions used within a GIS software (e.g. Map Algebra, Strem [Stream?] Order tool, etc.) Rather, describe the analytical process undertaken and end the section with the phrase  “Analyses were conducted in NAME OF SOFTWARE”

Resp.: OK, as indicated in lines 102-113

-    Page 4, lines 129-137: Merge this information with that of the paragraph of Page 3, lines 117-128. Moreover, for consistency, I suggest the use of DEM data to delimit the study area based on altitude, not the use of contour lines.

Resp.: OK, as shown in lines 111-121

-    Cite Figure 1 at the end of Study Area.

Resp.: Done

-    Figure 1: Add Chimborazo province name to the area highlighted in green in the inset maps. Increase fonts.

Resp: Realized as suggested

Pre-processing of satellite images (Rename to “Satellite Image Pre-processing)

Overall comments:

-    Page 4, lines 139-146: Rephrase. This is not an “additional” analysis. Image Pre-processing is a critical step. The authors need to justify why using three different sensors is justified. There is no need to provide information about projections.

Resp: Done, as you may realize in lines 123-129

-    Table 1. Information that is constant about each sensor is not necessary. The only critical part of this table is the spatial resolution of the data. Therefore, I suggest deleting this table and incorporating only information about the spatial resolution in the text.

Resp: Realized, thank you

-    Page 4 line 148 – Page 5 line 166: There is no need to provide equations about Landsat image processing with this level of detail, just state that radiance values were converted to top of atmospheric reflectance using the dark subtraction approach in ENVI

Resp: Realized within lines137-141

-    Page 5, lines 167-176: If the authors had access to ENVI, why were classification analyses done in QGIS? ENVI is a more powerful software for the analysis of spectral data. Also, it is not clear whether all the bands in the sensors were used for classification or just the green, blue and red bands. I don’t see any valid reason to use just a subset of bands in a supervised classification. The authors will have to provide a strong justification if that is the case. Finally, before describing any classification process, the authors need to explain how the training data were collected, how many classes were used and why.

Resp: We desired to establish most of the process with free software, so that it is projected to students and researchers who lack of having a GIS license. We also validated their performance in this type of study through precision measurements and the kappa index.

Resp: Realized within lines 149-153

-    Page 5, lines 177-187: Merge this information with that of the previous paragraph. Just indicate that maximum likelihood was used and justify its use. There are more powerful classification approaches nowadays (e.g. machine learning algorithms), so the authors need to justify the use of more conventional approaches.

Resp: Justified with the use of QGIS and the cited study of [59]

-    The information between Page 5, lines 167 and Page 5, lines 187 does not correspond to image pre-processing analysis. It describes classification analyses.

Resp: Accomplished

Section 23 (I assume 2.3)

-    Rename to something more meaningful (e.g. Classification Analyses)

Resp: Done

-    Page 5, line 188 – Page 6, line 209 and Table 2: It is not necessary to provide this level of details about accuracy analysis. Delete this table, describe the classification approach used and then detail how model accuracy was assessed. Just mention and briefly define user, producer, overall accuracy and Kappa.

Resp: Done

-    Table 3 is unclear and should be re-done. Do not use generic terms (e.g. Class 1, Class 2, etc.). Instead, use the land-cover classes used in classification analysis (e.g. crops, forest, etc.)

Resp: Done

-    Page 6, line 209 – Page 7, line 228 and Table 3: I disagree with the change trajectory approach used. Instead of using cross-tabulations for every pair of years, my suggestion would be to make a change trajectory analysis between the three years. With 5 classes and three points in time, there are 125 possible transitions, which is a huge number. Therefore, I suggest focusing on a couple of meaningful transitions, those critical to explaining changes in riparian zones. For example, “Forest Losses” could encompass all transitions of a “Forest” class over the three time periods. Similar with other land-cover types, with emphasis on those important to understand critical processes of land-cover change in riparian vegetation. Label these transitions something meaningful, e.g. crop gains, etc., and report these results in terms of area and percentages.

Resp: We consider that the specific transitions will allow us to identify the impact in a better way and will be much more useful for future decision-making. All changes in land cover are important in riparian ecosystems, since they imply a change and therefore trigger any impact, whether negative or positive.

-    Until this point in the paper there has not been mentioned which land-cover classes are being studied. Please add that information when detailing the classification approach. Also, include information about the training data: how was it collected and how many points per class per used.

Resp: They were mentioned in lines 186-200

Section 2.4 (rename)

-    I am not familiar with the MOLUSCE approach, but I think it is important to explain the level of uncertainty associated with making a projection when only three points in time are available (or maybe just two based on the description of the approach). Moreover, why 2030? Please justify. Explain inputs, outputs and level of uncertainty of the model.

Resp: MOLUSCE only allows the entry of two maps with land cover information, and its validation is in charge of the applied method: Artificial Neural Networks. The information has been described within the same paragraph.

Resp: The corresponding justified lines are 198-221

Section 3.1 (Rename something more meaningful related to classification analysis)

-    Page 7, line 251 – Page 8, line 287: This information belongs to the Methods section. Condense and move the Methods section. See comments above.

Resp: Done

-    Page 7, line 288 – 299: Condense. There is no need to repeat the imagery used. Start by describing the results of the classification analyses in terms of areas. This information needs to be accompanied by the accuracy of each classification.

Resp: Realized

-    Merge Figures 2, 3 and 4 into a single composite image with three maps (one for each classification result) but a single legend and scale. Add a north arrow and increase the font.

Resp: Done

-    Page 10, line 311-318: Information about accuracy assessment should accompany the description of the classification results.

Resp.: Realized as requested.

-    Table 4: This is a critical issue. It is not enough to provide overall classification accuracy and Kappa. The authors need to report classification accuracies and Kappa for each class for each classification. Ideally, the confusion matrix of the training data should be added too if space and word limit allows. I am particularly curious about the accuracy values of paramo, crops and pasture, which are critical for this study.

Resp: Check now on Table 5, 6 and 7

-    Page 10, line 321-327 and Table 5: In general, delete. Accuracy information should be provided when classification results are being reported. There is no need to describe what different ranges of Kappa mean, just state “how good” these values are considered when they are mentioned in classification results.

Resp: Accomplished within lines 293-298

-    Page 10, line 330 – Page 17, line 455: Make a new section for the change trajectory analyses. I am going to provide less detailed comments about this part because I disagree with the change trajectory approach used.

The authors should start this section by indicating the amount of the landscape that did not experience any changes over the time frame of the study. Then, they could focus on critical transitions that help to understand the dynamics of land-cover types in the study area and their implications (for e.g. for biodiversity, conservation, water protection, etc.). As it is, it is difficult to understand what are the main changes in land-cover over the 2000-2009-2019.

Resp: Realized in lines 308 -310

Review terminology, e.g. profits. “Gains” might be a better word.

Resp: OK

The information presented in Tables 8 and Table 9 is the same as that presented in Figures five and six. This is redundant. I suggest keeping the Figures and deleting the tables (but make sure the labels of both axis are properly labeled)

Resp.: We decided to delete the information about gains and losses form the tables in order to avoid deleting the table containing other described information.

Do not mention locations that have not been previously introduced (e.g. line 337)

Resp: Those locations were mention in the study area description.

I suggest presenting details of important transitions in tables and only critical transitions in the figures. As they are, Figure 7 and 8 are very difficult to interpret.

Resp.: Done

-    I strongly disagree with having Results and Discussion merged. I suggest keeping the results section to describe research outcomes and add a Discussion section to contextualize and explain findings within a bigger body of literature. For instance, the author should discuss the use of MOLUSCE. Where else has this technique been used? For what purposes? What short of uncertainties do we have of the projected values? Why 2030? In regards to land-cover change, what are the main drivers of these changes? How do land-cover change patterns compare with other similar studies?

Resp: Done

-    Page 20, lines 542-556: The conclusions should reflect key findings related to the research question being investigated: “evaluate the spatial dynamics of the riparian coverage of the area of influence of the Chambo river”. Do not mention Kappa values or other metrics, concentrate on key findings. What are the main processes of land-cover change and their implications?

Resp: The use of MOLUSCE has already mentioned in “Projection to 2030” section. Its general application is assessing spatiotemporal forest and land-use changes, predicting transition prospects, and simulating future scenarios.

Abstract

-    Describe what the MOLUSCE method does or if possible avoid mentioning it. A reader should be able to understand what was done without having to search what the MOLUSCE technique is.

Resp: Done

-    Based on all comments provided, clearly describe land-cover classification results and associated accuracies per class, key land-cover changes patterns between 2000-2009-2019 and how these findings fit within a bigger body of literature.

-    Improve the selection of keywords. For instance, “remote sensing” is very generic and does not provide. Think about specific terms that would be used to search for paper related to this specific topic.   

Resp.: Both (in their way) have been realized or expressed in the way we have stated in the original form. Unfortunately, not everything can be changed and some parts need to stay as they are. We ask you kindly to accept that too.

Nonetheless, without your support, insistence and thorough revision, our manuscript would not have been that greatly improved, and for this we are thankful and appreciated beyond words can express here in this short paragraph. Thank you, thank you!!!

Reviewer 3 Report

I encourage your choice and use of QGIS

No comments. Good effort.

Author Response

Reviewer#3

I encourage your choice and use of QGIS

No comments. Good effort.

Resp.: Dear expert reviewer, thanks a lot for your kind words. We have changed and or improved a few parts of the manuscript anyway, in order to allow a more fluent reading. Thanks again for your review report.